# AI Approaches to Environmental Impact Assessments (EIAs) in the Mining and Metals Sector Using AutoML and Bayesian Modeling

**Saki Gerassis** [1,*]📷, **Eduardo Giráldez** [1], **María Pazo-Rodríguez** [1], **Ángeles Saavedra** [2]📷 and **Javier Taboada** [1]

1   Research Group GESSMin, Department of Natural Resources and Environmental Engineering, University of Vigo, 36310 Pontevedra, Spain; egiraldez@uvigo.es (E.G.); maria.pazo@uvigo.es (M.P.-R.); jtaboada@uvigo.es (J.T.)
2   Research Group GESSMin, Department of Statistics and Operational Research, University of Vigo, 36310 Pontevedra, Spain; saavedra@uvigo.es
*   Correspondence: sakis@uvigo.es

**Abstract:** Mining engineers and environmental experts around the world still identify and evaluate environmental risks associated with mining activities using field-based, basic qualitative methods The main objective is to introduce an innovative AI-based approach for the construction of environmental impact assessment (EIA) indexes that statistically reflects and takes into account the relationships between the different environmental factors, finding relevant patterns in the data and minimizing the influence of human bias. For that, an AutoML process developed with Bayesian networks is applied to the construction of an interactive EIA index tool capable of assessing dynamically the potential environmental impacts of a slate mine in Galicia (Spain) surrounded by the Natura 2000 Network. The results obtained show the moderate environmental impact of the whole exploitation; however, the strong need to protect the environmental factors related to surface and subsurface runoff, species or soil degradation was identified, for which the information theory results point to a weight between 6 and 12 times greater than not influential variables.

**Keywords:** artificial intelligence; AutoML; Bayesian networks; sustainable mining; decision making; complex networks

## 1. Introduction

The mining and metals industry is one of the building blocks of the fourth industrial revolution. Raw materials are the indispensable foundation for a future digital society that relies on information and communication technologies at home and work, as well as for education and recreation [1–3]. Despite the increasing impact of circularity, recycling, reusing and reinvesting in raw materials will not be enough to meet consumer demands and industrial needs [4,5].

The challenge is enormous. Concretely, the 2020 Communication from the European Commission on Critical Raw Materials Resilience [6] indicates for batteries for electric vehicles and energy storage alone, up to 18 times more lithium and 5 times more cobalt will be required in 2030, and almost 60 times more lithium and 15 times more cobalt will be required in 2050, in relation to the current supply to the entire EU economy.

However, mining still poses serious and highly specific threats to biodiversity. Projections suggest that demand will grow for manifold mining metals such as cobalt, nickel or copper, shifting mining operations towards more dispersed and biodiverse areas [7,8]. In this scenario, mining companies urgently need to adapt their business models to become more sustainable, environmentally protective and resilient in the transition towards a low-carbon economy. For this reason, the sector is gearing up for innovation and technological transformation. Automation, robotics, machine learning, advanced analytics,

digital twin, predictive maintenance, the industrial internet of things (IIoT) and modern data architecture (including cloud computing) can help the industry to achieve this digital transformation [9–11].

Importantly, artificial intelligence (AI) could make the mining and metals industry more ethical and sustainable [12–14]. It may also help to reduce the industry's environmental footprint, moving workers out of mines, reducing costs and risks. At the same time, mining productivity can experience important gains, especially at a point in time where ore grades are declining, water scarcity has threatened to strand assets, and exploitation licenses have become extremely difficult to obtain.

Recent research and business developments show that achieving sustainable change requires more than a focus on technology [15–18]. In particular, one of the key challenges is to harness digital analytics by developing and implementing AI and machine learning models that effectively solve environmental, productivity and supply change problems. This study aims to deepen the manner in which environmental impact assessment (EIA) results are determined.

EIA is one of the principal tools used globally for sustainable mining [19–21]. It involves the application of different methodologies to evaluate the environmental impact of mining activities in the area of concern, supporting decision making ahead of new interventions. In essence, the EIA is a highly comprehensive tool that reflects adverse impacts in a concise manner, using typically an index [22,23]. The main advantage of the index is to gather information of manifold variables, which represent different environmental factors, evaluating the feasibility of the project and the possible need to implement mitigation measures. On the negative side, all these estimated outcomes are traditionally conditioned by mining engineers and environmental experts who evaluate the area of study and provide their qualitative and quantitative input to the index model.

In this study, the main goal is to introduce an innovative AI-based approach for the construction of EIA indexes that accurately reflects and takes into account the relationships between the different environmental factors, finding relevant patterns in the data and minimizing the influence of human bias. For that, this study takes advantage of the latest developments in automatic machine learning (AutoML) tools [24] merged with Bayesian modeling [25–27] to provide an agile user experience in the calculation and assessment of environmental impacts.

Practically, to carry out this study, a slate rock mine in the mining area of Quiroga in Galicia (north-western Spain) was chosen. Spain leads the world in producing and exporting natural slate, with a turnover in 2019 above 250 million euros, followed by China and Brazil [28,29]. The slate mines are located in the so-called Truchas syncline (between Ourense and León), which gave rise to "roofing slates", namely a type of metamorphic slate with a very fine grain and, in general, dark tones, which, due to its intense planar foliation, can be exploited to obtain plates for roofs, flooring and pieces of masonry.

Most of the existing mining designs are open-cast, with the slate layer being accessed by 4–8 m descending benches [30,31]. From an environmental viewpoint, this process involves large-scale removal of overburden and the subsequent creation of slate dumps in nearby areas. In the future, the continuity of these open-pit mines would be conditioned by two critical factors: first, the excellent environmental protection of the surroundings, especially in this case where the mining area of Quiroga has a privileged place under the Natura 2000 Network, given that more than 35% of the surface of this municipality is included in the Special Area of Conservation (SAC) and Special Protection Area (SPA) for Birds, according to the European Union's Directives on Habitats and Conservation of Wild Birds, respectively [32,33]; and second, the adequate transition towards underground mining operations as a means to maintain high mining extraction and productivity ratios.

This article, in line with the industry agenda developed by the World Economic Forum and United Nation's 2030 Sustainable Development Goals (SDGs) [34,35], describes the potential application of AI and AutoML to the mining and metals sector, outlining the key techniques and decision-making challenges that will shape its adoption in the transition

towards sustainable mining. Concretely, as introduced during this section, this research addresses the particular challenge of conceiving an EIA that, making use of AI and AutoML techniques, is capable of reflecting in a dynamic manner the adverse impacts of mining activity in the deterioration of ecosystems, including the pollution generated by secondary processes, such as natural slate manufacturing. Section 2 explains the methodology of carrying out a holistic analysis of the impacts, by automatically learning Bayesian networks from the data conceptualizing the environmental variables under study. Section 3 presents the results obtained, the algorithms applied, and the overall learning and cloud computing process put in place to identify those environmental factors with greater susceptibility to mining processes in the study area. Section 4 argues how AI-based platforms equipped with Bayesian inference techniques can help engineers and environmental experts to identify adverse environmental conditions, gaining insights from conceptual (human-based) and mathematical (simulation-based) modeling as a solution to better understand uncertainty. Finally, Section 5 draws the main conclusions.

## 2. Materials and Methods

### 2.1. Study Area

The slate reservoir is located in the town of Vilarbacú, in the mining area of Quiroga in Galicia (north-western Spain), which is designated as a UNESCO Global Geopark [36]. Figure 1 shows a map with the mine location, which is bordered to the east by a river basin (1.47 ha). Overall, the mine is surrounded by the Natura 2000 Network. In this geological domain, the rock mass corresponds to the Paleozoic metasedimentary that gives rise to different types of formations extremely rich in natural slate, covered by more recent superficial deposits which constitute the slate dumps resulting from the removal of waste material during the mining extraction and subsequent processing stages.

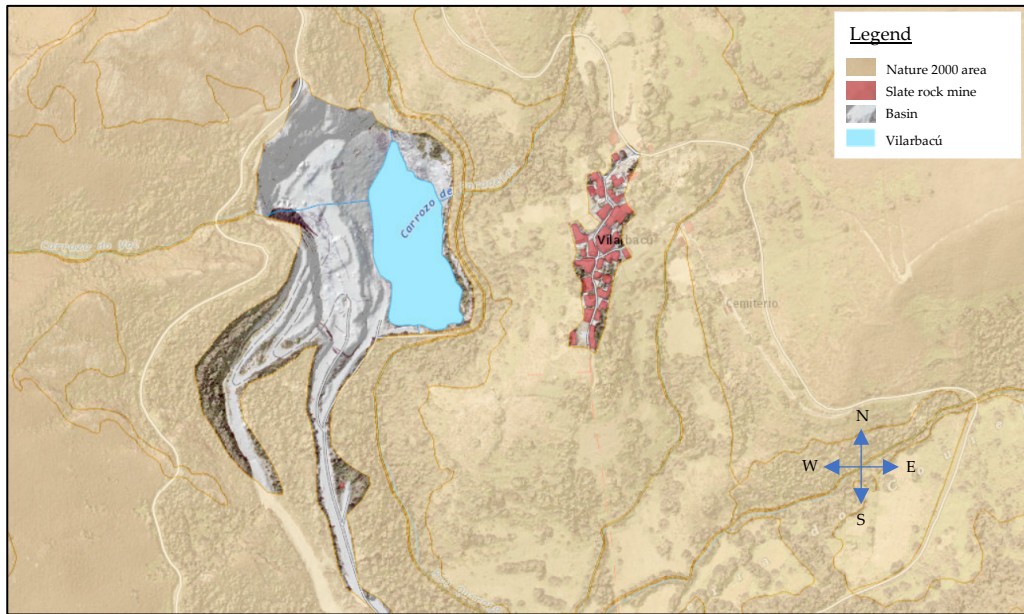

**Figure 1.** Map of the study area with the slate rock mine dominated by the Natura 2000 Network.

Given the importance of managing this outstanding geological site and, to promote sustainable research, the launch of the Natura 2000 Network increased the surveillance of the surroundings of the slate exploitation area, where extractive activities are concentrated. In consequence, the mining activity that was already taking place in the town of Vilarbacú was subject to rigorous EIA.

According to the characteristics of the terrain and the mining activity, it is reasonable to point to the fact that waste from the slate dumps exhibits the risk of negatively affecting the beds of nearby rivers, which are only a few kilometers away from the mining exploitation.

In this context, the release of relevant amounts of mineral powder directly into the terrain or through settling basins is the main pollution event to prevent. However, in these environments with a large number of endemic species (both flora and fauna), it is possible to identify numerous adverse effects that individually, cumulatively or simultaneously can create a state of imbalance in the environment [37,38].

### 2.2. Data for the Environmental Impact Assesment (EIA)

The EIA was conceived as an evaluation tool with the goal of identifying and assessing the environmental impact of an activity in advance [39]. As a result, the conclusions obtained from an EIA are intended to support decision-making regarding the implementation of policy or engineering interventions for environmental management. However, there exists an uncertain regulatory landscape in relation to the EIA methodological approach, particularly in the definition of a procedural application that can be both effective in pursuing the general objectives of environmental assessment and suitable for different environmental scenarios. In this context, there is a common understanding that an EIA needs to comply with the national and regional legislative framework of decision-making within which it operates [40].

Notwithstanding this rigidity, at this point, there is a great opportunity to develop modern data-driven approaches that put the focus on the data and the environmental factors and not so much on stringent criteria applied in one or another jurisdiction for the evaluation of impacts. On this basis, this research explores the development of an EIA that treats a priori all environmental factors equally, gathering the data on its corresponding attributes or variables which will constitute the input of an AutoML process, based on Bayesian learning, that will uncover the existing relationships and impacts among the different factors.

In total, 10 environmental factors were defined, which are characterized by a total of 40 attributes. Table 1 presents the 10 environmental factors clustered according to the potentially identified attributes that can be affected by the presence of mining activity in the study area of Vilarbacú. For each attribute, based on the analysis of the possible future environmental impacts associated with the mining activity evaluated in the field, a value between 0 (no impact) and 10 (huge impact) was assigned based on the degree of negative impact that this area may suffer as a result of the exploitation of slates. For each attribute, up to 60 different values were gathered in different locations within the study area.

Finally, it is important to remark that the scale of values assigned to the different attributes is intended to maintain the greatest possible similarity with the current narrative used in Spanish legislation [41] when categorizing the level of environmental impact. In practical terms, this does not affect the AutoML process proposed in this article; however, it may facilitate its interpretability by the public administration. Therefore, from a conceptual viewpoint, the attributes results are interpreted as follows:

- **Compatible environmental impact** (0–3): Recovery is immediate after the activity ceases, and no protection or mitigation measures are necessary.
- **Moderate environmental impact** (4–6): Protective measures or a recovery time interval are required after the activity ceases.
- **Severe-critical environmental impact** (7–10): It is necessary to apply protection and restoration measures, with the possibility that recovery after the cessation of activity is not possible given that its magnitude is above the acceptable threshold.

**Table 1.** Environmental factors and conceptual attributes associated with the study area.

| Environmental Factor | Attributes |
| --- | --- |
| Landscape Quality | Chromatic aspects; fragility; display (visualization); vulnerability |
| Soils | Water erosion, wind and desertification; soil degradation; surface and subsurface runoff; fertility decrease and soil recycling; edaphic profile changes |
| Ecosystems | Ecological succession break; biodiversity loss; food chain changes |
| Atmosphere | Noise; gases; dust; smell |
| Habitats | Construction of transport routes; roads construction; high voltage electricity grid; clearing and leveling |
| Surface Waters | Ecosystem loss, eutrophication and water quality; water pollution by waste dump; flow changes; basin contribution changes; basin edaphology changes; current flow changes; flood zones alteration |
| Fauna | Fauna biodiversity and habitat loss; fauna unprotected species decrease; isolation of species or individuals; species or individuals concentration |
| Flora | Flora biodiversity and habitat loss; flora unprotected species decrease; decrease in flora growth and regeneration |
| Morphology | Shapes and volumes; slopes changes |
| Geophysical Processes | Erosion changes; slopes stability alteration; vibrations; deposition |

*2.3. AutoML for Understanding the Environmental Impact Assesment (EIA)*

Automating the creation of EIAs with AutoML through advanced digital tools opens the door to create a more robust decision-making framework for the assessment of environmental impacts, with the possibility of easily involving engineers and environmental experts with limited knowledge on programming tasks. Therefore, by using the click-and-point convenience of AI-based cloud platforms, it is possible to redefine the creation and interpretation of EIAs.

The methodology proposed aims at automating the entire data pipeline, laying down the foundations for a dynamic tool capable of updating the potential environmental risks in the study area as soon as new information or expert evidence are introduced in the model. To conduct the AutoML process, it is necessary to select a learning method. In this approach, the authors endorse Bayesian networks as the learning paradigm. The rationale for using Bayesian networks is justified in their prediction accuracy and excellent graphical capabilities [42]. Altogether, Bayesian networks are an excellent tool to simultaneously represent a large set of probabilistic relationships, involving decision makers to test their beliefs.

More precisely, a Bayesian network is defined as a directed acyclic graph (DAG) that represents a set of attributes together with their conditional dependencies. Formally, if $\Omega = \{X_1, X_2, \ldots, X_n\}$ is a set of variables, in consequence, a Bayesian network for $\Omega$ is defined as a pair <G, P>, where:

- G is a DAG in which each node represents one of the variables $X_1, X_2, \ldots, X_n$ and, each arc represents a direct relationship of dependency between variables;
- P is a set of parameters that typify the network by reflecting the probabilities for each possible value $x_i$ of each variable $X_i$.

When building Bayesian networks using AutoML, the learning process becomes an automatic data-driven process governed by algorithms with specific learning characteristics. The following subsections will detail the specific steps and Bayesian algorithms performed to build the model.

To create the whole AutoML process the AI platform BayesiaLab v9 [43] was used. BayesiaLab is a software as a service (SaaS) that provides a laboratory environment for

knowledge modeling with Bayesian networks, complemented with a powerful graphical user interface that allows to explore the causal directions in the network graph.

### 2.3.1. Data Discretization

As a first step, the data introduced in the AutoML process need to be discretized. Discretization has a critical impact on the model because it determines the characteristics of the domain to be modeled. In this case, all the attributes are continuous variables with a value between 1 and 10. Importantly, this aspect should not be confused with the conceptual discretization for subsequent environmental interpretation purposes (Section 2.2).

One of the most important factors to take into consideration when carrying out discretization is the number of states or intervals within an attribute. It has a direct impact on the model's complexity. Typically, the higher the number of states, the more complex the model. However, the volume of data available must be considered. Concretely, with small volumes of data, it is safer to choose a conservative number of states that concentrate the informational content, giving rise to significant arc relationships in the Bayesian networks to be generated.

In this case, 60 observations per attribute can be considered a small to medium volume of data, regardless of the overall dataset volume. Therefore, 3 states per attribute were chosen. To execute the discretization, the algorithm K-means was used. BayesiaLab v9 provides a set of discretization algorithms that cover different applications depending on the data under study. In this case, K-means was the preferred option given its reliability in multiple scenarios, based on the expectation-maximization (EM) approach, where from a random creation of K centers, each point is linked with the closest center and the position of each center is computed as the barycenter of its associated points.

### 2.3.2. Unsupervised Bayesian Learning

Learning a Bayesian network from an optimal classifier is a NP-hard problem [44]. Generally, learning the typological structure of the network is a delicate and frequently time-consuming step. In this case, an unsupervised learning approach is applied, which represents the best option to machine learn the a priori unknown structure and interrelationships from the data in the attributes.

The minimum description length (MDL) score was selected as the score-based learning function to assess the quality of the candidate Bayesian network with regard to the data describing the mining and energy problems under analysis. The MDL principle is derived from information theory and formalizes the fact that the best explanation for a given set of data is provided by the shortest description of that data.

Mathematically speaking, the MDL is a two-component score, where the model is the Bayesian network (graph and probability tables) and the data given to the Bayesian network are inversely proportional to the probability of the observations returned by the model. Formally, it can be defined as:

$$\text{MDL (Bn, D)} = \alpha \text{DL(Bn)} + \text{DL(D|Bn)} \tag{1}$$

where DL(Bn) represents the complexity of the network and constitutes the number of bits required to represent the Bayesian network (Bn), and DL(D|Bn) is the fit of the network and implies the number of bits required to represent the dataset D given the Bayesian network (Bn). Finally, $\alpha$ is the structural coefficient of the network [43]. This is a critical parameter which reflects the strength of the probabilistic relationships needed to result in a network arc.

In order to analyze the structural learning results from the algorithms applied and to select the candidate networks, the objective is to minimize the MDL score. In practical terms, minimizing the MDL score consists of finding the best trade-off between complexity and data representation [45]. In this respect, a key aspect for decision makers when using AutoML with Bayesian networks will be to adjust $\alpha$ in a manner that accurately reflects

the number of available observations. In consequence, the correct estimation of $\alpha$ for the network will allow researchers to validate its application in the study.

### 2.3.3. Computing Risk and Uncertainty

One of the advantages of using an AutoML with Bayesian networks is the possibility of applying information theory to introduce a sophisticated treatment of uncertainty. As a starting point, the Shannon entropy H(X) of a random variable allows one to quantify the uncertainty associated with the probability distribution of a variable X or a set of variables $\Omega$.

$$H(X) = -\sum_{x \epsilon X} p(x_i) \log_2(p(x_i)) \tag{2}$$

where $x_i, \ldots, x_n$ are the possible outcomes of X, which occur with a probability $p(x_i), \ldots, p(x_n)$, representing the complexity of the network and constituting the number of bits required to represent. Full or maximum uncertainty $H_{max}$ occurs when all possible states of a node are equally probable. In fact, the maximum value of entropy increases logarithmically with the number of states $\phi_x$ of a variable X.

$$H_{max}(X) = \log_2(\phi_x) \tag{3}$$

As a result, the more possible states in a node, the maximum entropy increases [46]. Importantly, evaluating the degree of uncertainty of the whole EIA requires one to calibrate the limits between no uncertainty and maximum uncertainty. The former is zero, while the latter can be obtained by applying Equation (3), considering the different attributes connected in the network. To make it simpler, it is possible to calculate a normalized entropy that can directly provide the value of uncertainty in a percentage manner.

$$H_n(X) = \frac{H(X)}{\log_2(\phi_x)} \tag{4}$$

Lastly, it is important to highlight the possibility of integrating these information theory parameters in an easy manner in the network structure through function nodes. A function node, illustrated with a hexagon, allows one to introduce a particular mathematical equation that is estimated based on the values of the attributes in the network.

## 3. Results

### 3.1. Exploratory Analysis of the Unsupervised Network

From the construction of an unsupervised Bayesian network, the potential relationships between variables can be explored in reality, transferring them to the model [23]. In this manner, it is possible to carry out a global analysis of the problem, detect which nodes have the greatest influence and obtain an understanding of the individual influence of the variables under study. An overview of EIA is shown in Figure 2.

The winning network model, which best represents the field under study, was built using the EQ algorithm in BayesiaLab v.9 [43]. This learning method explores the space of equivalence classes of Bayesian network structures. This method is highly efficient because it reduces the size of the search space to partially directed acyclic graphs (PDAGs), smaller than the space of Bayesian networks (DAGs), in order to represent the equivalence classes evaluated during each search, directly calculating their score. A comprehensive algorithm portfolio, including other relevant algorithms such as maximum weight spanning tree (MWST) or taboo were tested. However, the lowest minimum description length (MDL) value was obtained with EQ, indicating the best trade-off between complexity and data representation (Section 2.3.2) and validating its adoption in this study.

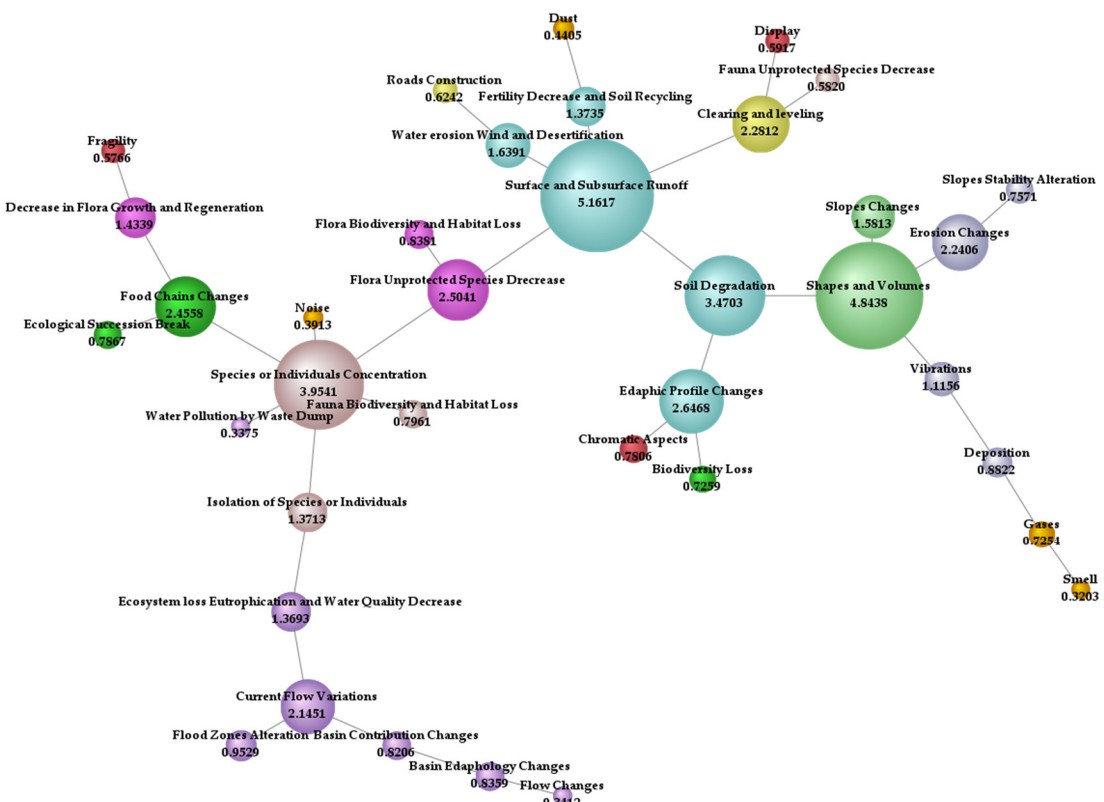

**Figure 2.** Unsupervised Bayesian network using node force mapping for the complete EIA representation. The color of the nodes outlines a conceptual clustering (human-based).

The attributes in Figure 2 are highlighted by colors using expert criteria according to the aspect or type of species affected by the presence of mining activity in the study area. By making a comparison between this conceptual cluster (human-based) in Figure 2 and the statistical cluster (model-based) in Figure 3 derived from the network, it is possible to identify relationships between variables that initially belong to other clusters. The conceptual attributes belonging to the environmental factors *Habitats*, *Surface Waters*, *Morphology*, *Flora* and *Geophysical Processes* (Table 1) maintain their association in the Bayesian network. However, in the Bayesian model created from the attributes for this EIA, it is observed that not all the factors that were conceptually believed to be related have a proven relationship in reality. Note the absence of correlation between the attributes *Fragility*, *Chromatic Aspects* and *Display*, which reflects the fact that the heuristics used by the EQ algorithm did not find any relevant causal relationship between the factors that belonged to the *Landscape Quality* conceptual cluster.

Based on the arc interrelationships and the numerical values associated with each variable, it is observed that *Surface and Subsurface Runoff*, *Shapes and Volumes*, *Species or Individual Concentration* and *Soil Degradation* are potentially the four nodes with the greatest impact on the affected natural space.

For instance, *Species or Individual Concentration* proves to be a determining factor on the characteristics of surface waters: *Ecosystem Loss, Eutrophication and Water Quality Decrease* → *Current Flow Variations* → *Basin Contribution Changes* → *Basin Edaphology Changes* → *Flow Changes*. An interesting aspect to highlight after calculating the strength of the nodes is that the four attributes identified have a weight between 6 and 12 times greater than the only two attributes (*High Voltage Electricity Grid* and *Vulnerability*) that are not part of the network due to their low capability to establish significant connections with other nodes.

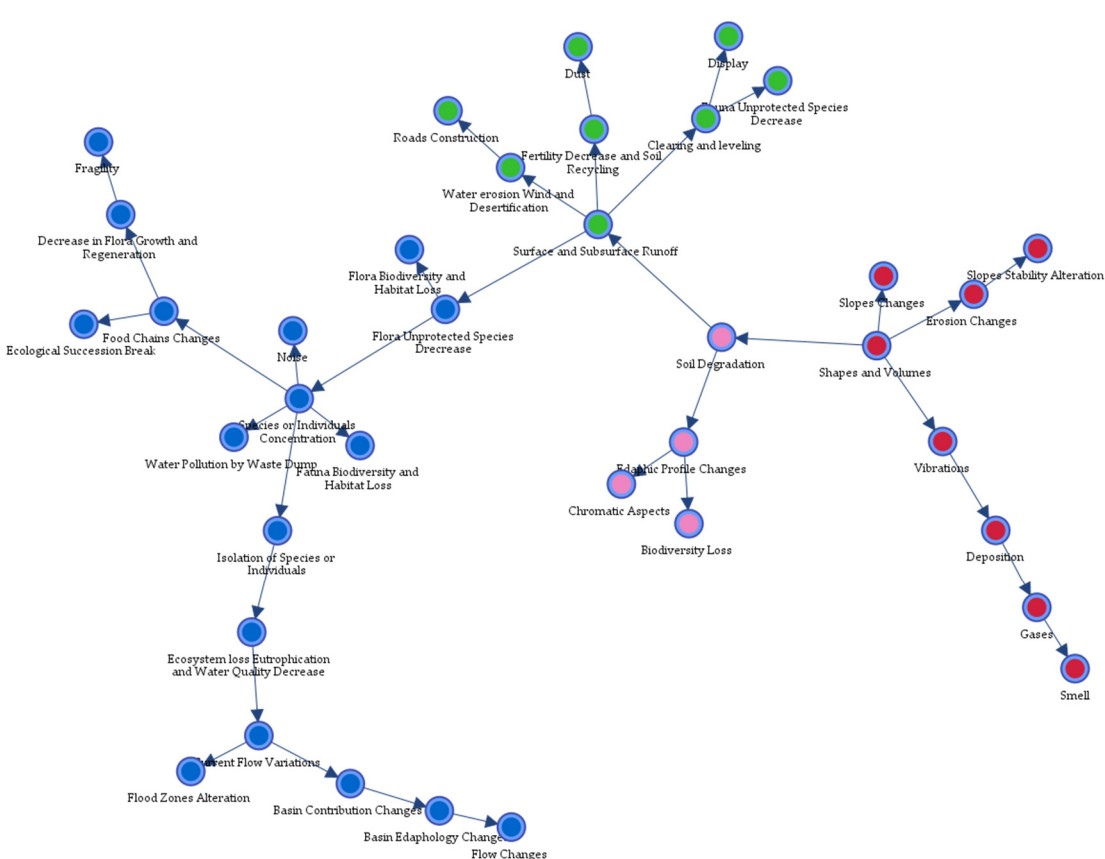

**Figure 3.** Unsupervised Bayesian network built with the EQ algorithm divided into mathematics clusters. The color of the nodes outlines a statistical network clustering (model-based).

From the exploratory analysis of the unsupervised network, it is possible to identify the cause-effect relationships between variables. This great potential of the Bayesian network automatically learnt from the data allows us to observe that, in terms of the severity of the EIA, the appearance of *Surface and Subsurface Runoff* is closely related to *Soil Degradation*, which in turn is related to possible modifications or alterations that the mining activity causes on the forms and volumes of the land.

The previous factor seems to play a fundamental role in the problem to be analyzed, acting as the starting point of all the dependency relationships of the network (Figure 3), which gives it great conceptual weight. In the first place, it is reasonable to assume that changes in the morphology of the terrain affect the rest of the environmental factors in the network, since it is the basis for the life of flora and fauna, and supports the entire ecosystem. However, for its justification, the level of dependence of these variables with the *Shapes and Volumes* factor was evaluated, including the impact it generates on the network according to its degree of alteration by means of an inference analysis (Section 3.2).

In relation to this analysis, the mutual information (MI) between the nodes was calculated to find out which are the attribute relationships that provide the greatest gain of information to the EIA. This information theory measure allowed for identifying that *Shapes and Volumes* in addition to *Slope Changes* are the attributes with the greatest conceptual weight, having the strongest correlations (Pearson correlation) in the network with their peers (Table 2).

**Table 2.** Relationship analysis.

| Parents | Child | Relative Weight | Contrib. | MI | PC |
|---|---|---|---|---|---|
| Shapes and Volumes | Slopes Changes | 1.0000 | 5.78% | 1.5813 | 0.9999 |
| Shapes and Volumes | Erosion Changes | 0.9381 | 5.42% | 1.4835 | 0.9860 |
| Soil Degradation | Surface and Subsurface Runoff | 0.7521 | 4.35% | 1.1894 | 0.9752 |
| Shapes and Volumes | Soil Degradation | 0.7213 | 4.17% | 1.1406 | 0.9170 |
| Soil Degradation | Edaphic Profile Changes | 0.7211 | 4.17% | 1.1403 | 0.9706 |
| Surface and Subsurface Runoff | Clearing and leveling | 0.7004 | 4.05% | 1.1075 | 0.9624 |
| Surface and Subsurface Runoff | Water erosion Wind and Desertification | 0.6419 | 3.71% | 1.0150 | 0.9150 |
| Current Flow Variations | Flood Zones Alteration | 0.6026 | 3.48% | 0.9529 | 0.9982 |
| Surface and Subsurface Runoff | Fertility Decrease and Soil Recycling | 0.5900 | 3.41% | 0.9330 | 0.9499 |
| Surface and Subsurface Runoff | Flora Unprotected Species Decrease | 0.5798 | 3.35% | 0.9169 | 0.9127 |

### 3.2. Predicting the Environmental Impact Assessment (EIA) Index

In relation to the exposed problem, and considering the need to determine an EIA index, it was decided to analyze various scenarios from the inference point of view (Figure 4). From a more general perspective, the global average value of the network obtained from the sum of the probabilistic values of the nodes is 3.839, which corresponds to a moderate environmental impact. This value is obtained with the function node 'mean value' in Figure 4.

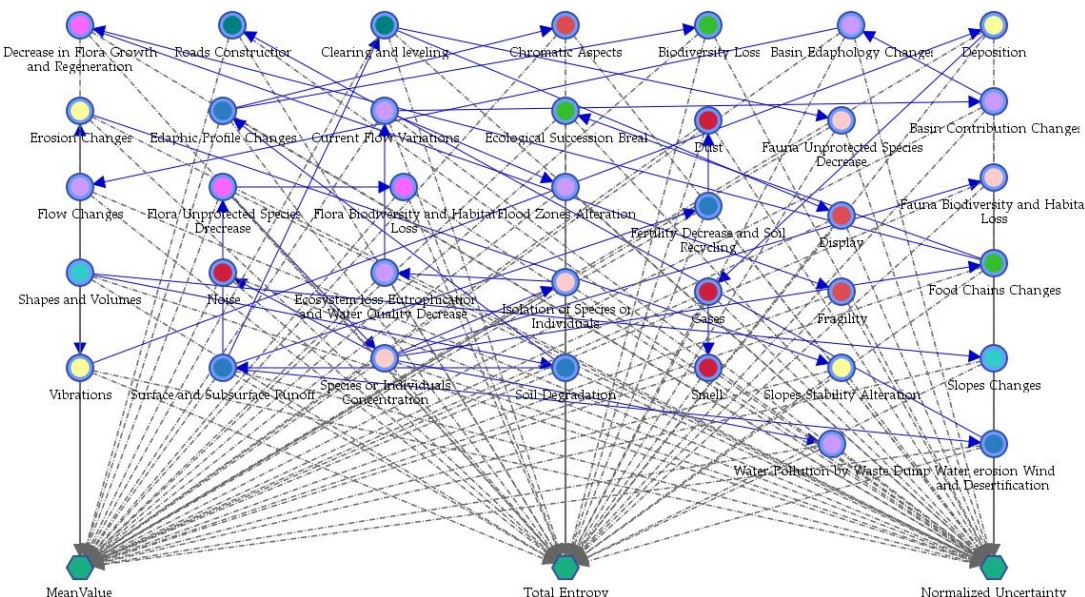

**Figure 4.** Function node tool (green hexagon) for mean value, uncertainty [%] and entropy [bits] calculation.

More specifically, the environmental factor with the greatest impact according to the percentage distribution of the network is the flora biodiversity and habitat loss; its frequency indicated that in 66.67% of the cases, it will be in a state of severe-critical environmental impact (6.432–10), followed by soil fertility and recovery (65% (5.256–10)) and unprotected flora species decrease (63.33% (6.202–10)). Within this approach, the variables with the greatest weight identified in the unsupervised network (Figure 2) did not show notable differences from the final values of the EIA (Table 3). However, the uncertainty associated with its probability distribution, for low values of affection, can be reduced by up to 50.69%. In this sense, it was observed that if these four factors remain in a low state of alteration, 87% of the remaining cases would not need restoration work, simplifying the activities related to the cessation of mining activity at the end of the life of the mine

**Table 3.** Inference results.

| Intervals | | Surface and Subsurface Runoff | Shapes and Volumes | Species or Individual Concentration | Soil Degradation |
|---|---|---|---|---|---|
| Compatible | Mean Value | 1.308 | 1.308 | 1.078 | 1.308 |
| | Uncertainty [%] | 11.678 | 11.678 | 8.66 | 11.678 |
| | Entropy [bits] | 18.510 | 18.510 | 13.726 | 18.510 |
| Moderate | Mean Value | 3.593 | 4.318 | 4.447 | 3.939 |
| | Uncertainty [%] | 16.915 | 18.092 | 21.502 | 18.291 |
| | Entropy [bits] | 26.809 | 28.674 | 33.798 | 28.990 |
| Severe-Critical | Mean Value | 5.236 | 5.529 | 5.780 | 5.317 |
| | Uncertainty [%] | 16.964 | 14.607 | 18.517 | 16.251 |
| | Entropy [bits] | 26.536 | 22.689 | 29.02 | 25.378 |
| | | Mean Value | Uncertainty [%] | Entropy [bits] | |
| Total | | 3.839 | 27.079 | 42.677 | |

During this network analysis, it was also detected that the conceptual cluster for *Surface Waters* (Figure 5) maintained a close relationship with the analytical grouping. Note, in the representation of Figure 6, the great variation of the cluster percentage distributions under a possible moderate (2.109–7) or severe-critical (7–10) degree of alteration for the *Basin Contributions Changes*. In fact, it is demonstrated that any alteration that implies the instantaneous non-recovery of the characteristics of the basin would imply the need to intervene and establish protection measures, or in the case of critical conditions, restoration measures.

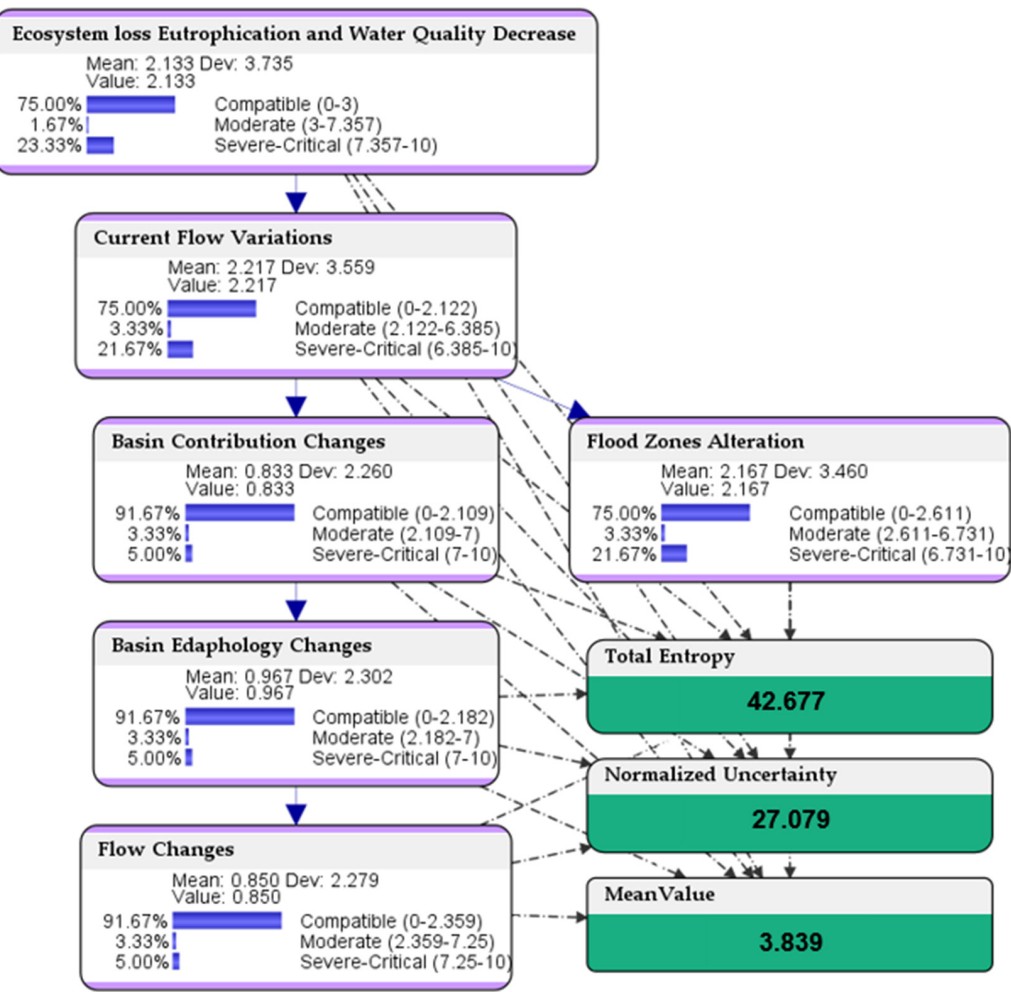

**Figure 5.** A priori probabilities for the variables before conducting inference analysis.

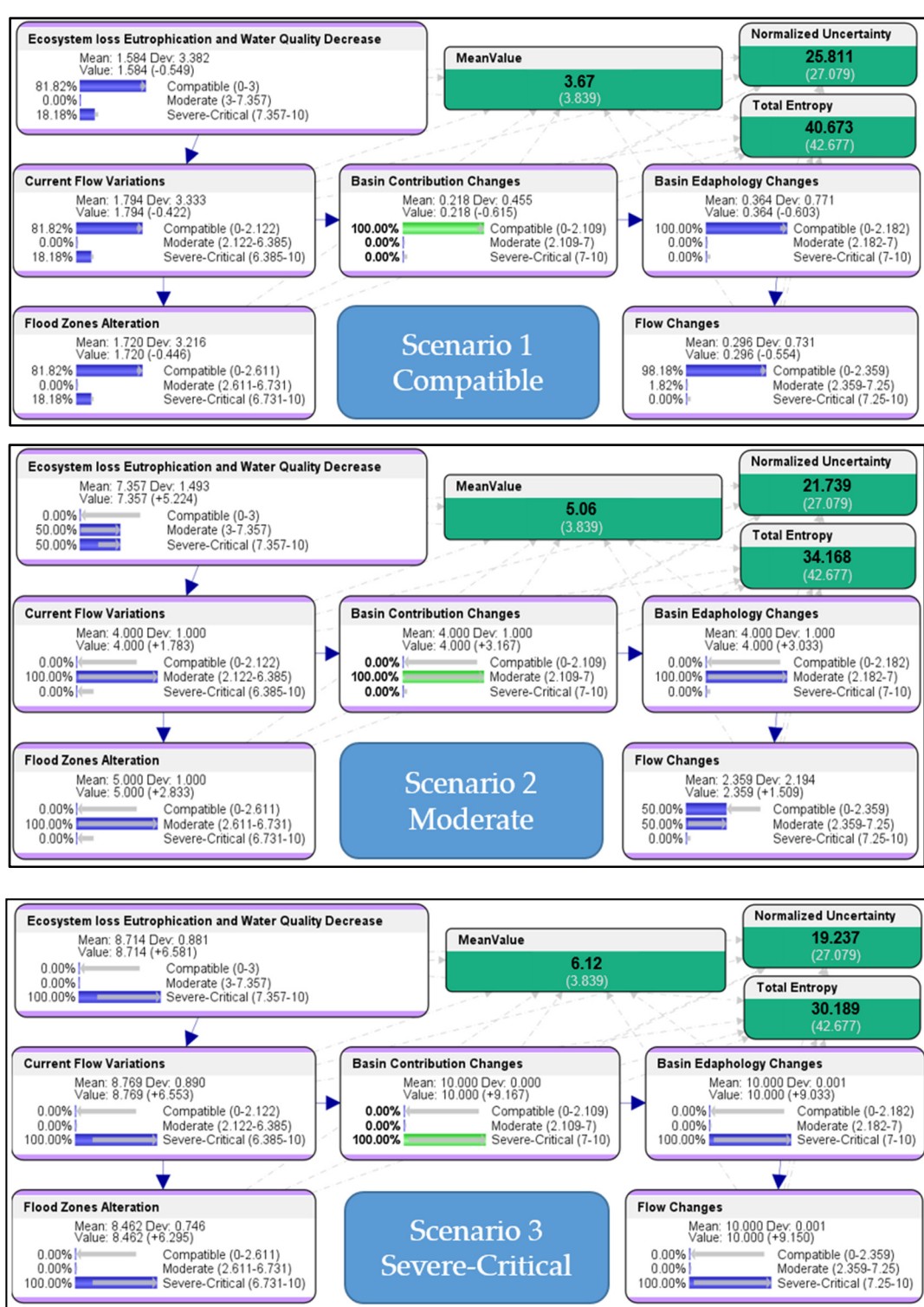

**Figure 6.** Inference analysis of the *Basin Contribution Changes* attribute. Three scenarios: 1—compatible, 2—moderate and 3—severe-critical.

Within this same scenario, it was identified from an influence perspective on the mean EIA index of the entire network that if major changes take place in the edaphology (7–10), flow (7.25–10) and contribution (7–10) to the basin, the EIA index rises to a medium-high value (6.12), decreasing its uncertainty by 8%. Statistically, this result implies that 83.78% of the evaluated factors would need protection or mitigation measures if a periodic control was be carried out.

## 4. Discussion

In the present case study for a natural slate mine in the mining region of Quiroga (Spain), an information theory analysis based on unsupervised Bayesian network allowed for identifying the factors *Surface and Subsurface Runoff*, *Shapes and Volumes*, *Species or Individual Concentration* and *Soil Degradation* as the possible nodes with the greatest environmental impact on the study area. In order to facilitate decision-making in situations in which there is uncertainty, using Bayesian inference techniques, a medium-low environmental impact index (3.839) was obtained with a degree of uncertainty equal to 27.079% (Figure 5).

Likewise, once the conceptual and mathematical relationships associated with the *Surface Waters* cluster were identified, three study scenarios were assessed based on the level of alteration of the contributions made to the basin. Through Bayesian inferential analysis, it was possible to better understand the uncertainty and observe the great variation in the percentage distributions of the environmental factors of interest related to this factor.

In Figure 6 are presented three scenarios (1—compatible, 2—moderate and 3—severe-critical) for which the variation in *Basin Contribution Changes* notably affects the environmental conditions of other attributes. For example, it is surprising how *Flood Zones Alteration* or *Current Flow Variation* are completely coupled to *Basin Contribution Changes*, presenting the same results. However, *Flow Changes* shows a more uncertain behavior, where in scenario 2 shows a 50% between compatible and moderate.

In this context, it is fundamental to remark that this inference analysis corresponds to a relatively small part of the whole network (Figures 2 and 3). The analysis possibilities are almost limitless and depend on a good understanding of the study. In this respect, it is recognized how water may influence the environmental risk of natural slate mines [31,37].

This study shows the great potential of AutoML and Bayesian networks to reason under uncertainty, bringing flexibility to decision making. In particular, the results prove the value of information (VoI) obtained with a model-based simulation that excels simple human-based approaches that are still widely used nowadays [22,23,47]. Therefore, it is possible to achieve statistical evidence of the heterogeneity of the circumstances established in the definition of the EIA index. In the coming years, the implementations of AI-based platform solutions as part of the overall digitalization of the mining and metals industry should contribute to the development of sustainable mining.

Initiating dialogue between mining companies, policy makers and environmental organizations is urgent [34,48]. Areas of special natural interest, such as the one presented in this study, where a mine is completely surrounded by the Natura 2000 Network (Figure 1), are not an exception. In fact, in the future, these types of situations are only expected to grow in prevalence due to the increasing policy activity of legislators on biodiversity. Undoubtedly, this is a positive aspect. At the same time, mining activity is necessary and should be further stimulated in a social context where raw materials demand increases relentlessly to meet consumer and industrial needs. This article shows how a new technological paradigm supported by AI can scientifically help demonstrate or debunk the negative impact of mining activity, defeating lobbies and interested groups and bringing finally certainty to a sector, which has traditionally suffered from volatile geopolitical events.

## 5. Conclusions

To conclude, mining activity for the extraction of natural slate has historically been denoted by society as a strongly aggressive exercise for the environment due to the large amount of waste generated. On the other hand, it is important to note the need that society has for this type of raw material, and mining in general. In this respect, the authors firmly believe that it is possible to conduct mining in a sustainable manner supported by digitalization and the pervasive adoption of AI tools that support informed decision making.

**Author Contributions:** Conceptualization, S.G.; methodology, S.G., E.G., Á.S. and J.T.; software, M.P.-R. and S.G.; validation, S.G., E.G., M.P.-R., Á.S. and J.T.; formal analysis, Á.S. and M.P.-R.; investigation, E.G.; resources, E.G., Á.S. and J.T.; data curation, S.G., M.P.-R., Á.S. and J.T.; writing—original draft preparation, S.G., M.P.-R. and J.T.; writing—review and editing, E.G. and Á.S.; visualization, M.P.-R. and J.T.; supervision, S.G. and J.T.; project administration, J.T. All authors have read and agreed to the published version of the manuscript.

**Funding:** This research received no external funding.

**Institutional Review Board Statement:** Not applicable.

**Informed Consent Statement:** Not applicable.

**Conflicts of Interest:** The authors declare no conflict of interest.

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
