# Peer review of "AI Approaches to Environmental Impact Assessments (EIAs) in the Mining and Metals Sector Using AutoML and Bayesian Modeling"

_applsci, doi:10.3390/app11177914_

Round 1
Reviewer 1 Report
The article verses on the enviromental impact of mining, and shows the potential of Bayesian Networks (and other IA and Machine Learning techniques) to access impact.
The paper is well-written, sound and I concur with publication.
In my opnion, the paper can be published as is. However, if the authors are interested, I would like to say that the submitted article uses entropy and complex networks, while its set of references is completely detached from the broad range of publications relating the topics.
The most important article is:
https://doi.org/10.1103/PhysRevE.70.066117
But here are more recent references:
https://doi.org/10.1088/2632-072X/ab9447
https://doi.org/10.22191/nejcs/vol3/iss1/5
https://arxiv.org/abs/2106.13565
https://arxiv.org/abs/2106.14784
Author Response
Answer in the document attached

Reviewer 2 Report
The authors present an interesting and novel approach for conducting Environmental Impact Assessments in the mining and metal extraction sectors using modeling approaches. Overall, the paper provides significant information that could have wide applicability in improving and automating the existing EIA methodologies in these industrial sectors. However, the article requires significant language editing since the current presentation is slightly disjointed and occasionally unclear in terms of the language and grammar used. Some examples are listed below (not comprehensive):
- Title:
Current: Redefining Environmental Impact Assessments (EIAs) in the
mining and metals sector using AutoML and Bayesian
modeling
Recommended: Innovative AI approaches to Environmental Impact Assessments (EIAs) in the mining and metals sector using AutoML and Bayesian modeling
2. Line 12: " Mining engineers and environmental experts around the world still today identify and evaluate the environmental risks associated to mining activities from rudimentary qualitative methods based on field checks"
Recommended: "Mining engineers and environmental experts around the world still identify and evaluate environmental risks associated with mining activities using field-based, basic qualitative methods"
3. Line 20: "The results obtained allowed to conclude"
Recommended: "The results obtained showed.."
4. Line 31 "at home, work, education and recreation"
Recommended: at can be used in the context of home and work but "for" is more appropriate for education and recreation.
5. Line 32: " in the next years"
Recommended: delete those 4 words
6. Line 35: "Concretely, the 2020 Communication from the European 35
Commission on Critical Raw Materials Resilience [6] indicates that only for batteries for electric vehicles and energy storage, it will be needed up to 18 times more lithium and 5 times more cobalt in 2030, and almost 60 times more lithium and 15 times more cobalt in 2050, in relation to the current supply to the entire EU economy"
Recommended: Please rephrase since the use of "it will be needed" is not correct
7. Line 41: Clarify what is meant by "manifold metals". Does this refer to piping?
8. Line 42: Rearrange "need urgently to" to "urgently need to"
9. Line 45: "Automation and robotics, machine learning and advanced analytics, digital twin and predictive maintenance, the Industrial Internet of Things (IIoT), and modern data architecture (including cloud computing) can help the industry to achieve this digital transformation"
Check the use of commas & "ands" in the sentence above.
10. Line 60: "aims to deepen into this issue" is not correct grammar
11. Line 62: Remove "The" from The EIA
12. Line 67: Replace "concluding" with "evaluating"
13. Line 69: Please clarify what is meant by "on the negative side"
14: Line 78: Remove "over traditional methodologies largely used for many years"
15: Line 79: "to carry out this study it was chosen a slate rock mine in the mining area" change to" to carry out this study, a slate rock mine in the mining area was chosen"
16. Line 80: "Spain leads the world ranking of countries producing"
change to " Spain leads the world in producing..."
17. Line 103: Remove "a digital and"
18. Line 122: Please correct "which is in direct contact by the East"
19. Line 129: Consider rephrasing "the entry into force of the action plan"
20. Line 135: Consider rephrasing "being the closest few kilometers from the mining exploitation"
21. Line 136: Please clarify what is meant by "mineral power"
22. Line 139: Replace "locate" by "identify"
23. Line 140: Replace "generate" by "create"
24. In Figure 1, please explain the legend entry marked "Habitats Plan Director RN2000"
25. Line 147: "activity or specific action" sounds incorrect
26. Line 150: Replace "uncertainty" with "uncertain"
27. Line 178: In table 1, what is meant by "Display" as an attribute to Landscape quality. Does this refer to Visual?
28. Line 217: Please correct "To conduct the AutoML process is necessary to select a learning method"
29. Line 219: Please correct "accuracy joint to their excellent..."
30. Line 233: Please correct "will deepen on the specific steps"
31.Line 246: Please correct "important factors to have into consideration"
32. Line 254: Please correct "Therefore, it was chosen a number of 3 states per attribute"
33. Line 300: Please correct "allows to quantifies the uncertainty"
34. Line 336: What is meant by "The winning network model"?
35. Line 406: The number "3,839" seems incorrect and should be changed to 3.839
36. Line 410: Figure 4 can be improved in clarity since it is currently not very readable
37, Line 415: The number "6,432" is incorrect and should be changed to 6.432
38. Line 422: Please clarify "it would be avoided to take the risk that after..."
39. Line 441: The quality of Figure 5 should be improved, several parts are currently faded
40. Line 450: Check the number 3,839
41. Line 464: Correct "good understanding of the study are..."
42: Line 468: The quality of Figure 6 can be improved, particularly the bracketed values, 3839 for example
43: Line 468: Check numerical values displayed in Figure 6. For example, the Mean Value shown is 3,67 instead of 3.67. This is repeated several times in different areas of the figure.
44: Line 472: Please correct this line " that excels simple human-based approaches, still widely used nowadays"
45: Line 477: Please correct "contribute to set in a real sustainable mining"
46: Line 489: Consider rephrasing for clarity: "To conclude, the mining activity for the extraction of natural slate has been denoted by society to a great extent, as a strongly aggressive exercise for the environment due to the large amount of waste generated."
Author Response
Answer in the document attached
